# Persistent cross-species transmission systems dominate Shiga toxin-producing *Escherichia coli* O157:H7 epidemiology in a high incidence region: A genomic epidemiology study

Gillian AM Tarr[1]*, Linda Chui[2,3], Kim Stanford[4], Emmanuel W Bumunang[4], Rahat Zaheer[5], Vincent Li[2], Stephen B Freedman[6], Chad R Laing[7], Tim A McAllister[5]

[1]Division of Environmental Health Sciences, School of Public Health, University of Minnesota, Minneapolis, United States; [2]Alberta Precision Laboratories, Alberta Public Health, Walter Mackenzie Health Sciences Centre, Edmonton, Canada; [3]Department of Laboratory Medicine and Pathology, University of Alberta, Edmonton, Canada; [4]Department of Biological Sciences, University of Lethbridge, Lethbridge, Canada; [5]Agriculture and Agri-Food Canada, Lethbridge Research and Development Centre, Lethbridge, Canada; [6]Sections of Pediatric Emergency Medicine and Gastroenterology, Department of Pediatrics, Alberta Children's Hospital and Alberta Children's Hospital Research Institute, Cumming School of Medicine, University of Calgary, Calgary, Canada; [7]National Center for Animal Diseases Lethbridge Laboratory, Canadian Food Inspection Agency, Lethbridge County, Canada

*For correspondence: gtarr@umn.edu

## eLife Assessment

This **valuable** study revealed numerous distinct lineages that evolved within a local human population in Alberta, Canada, leading to persistent cases of *E. coli* O157:H7 infections for over a decade and highlighting the ongoing involvement of local cattle in disease transmission, as well as the possibility of intermediate hosts and environmental reservoirs. This study also showed a shift towards more virulent stx2a-only strains becoming predominant in the local lineages. The evidence supporting the role played by cattle in the transmission system of human cases of *E. coli* O157:H7 in Alberta is **solid**.

**Abstract** Several areas of the world suffer a notably high incidence of Shiga toxin-producing *Escherichia coli*. To assess the impact of persistent cross-species transmission systems on the epidemiology of *E. coli* O157:H7 in Alberta, Canada, we sequenced and assembled *E. coli* O157:H7 isolates originating from collocated cattle and human populations, 2007–2015. We constructed a timed phylogeny using BEAST2 using a structured coalescent model. We then extended the tree with human isolates through 2019 to assess the long-term disease impact of locally persistent lineages. During 2007–2015, we estimated that 88.5% of human lineages arose from cattle lineages. We identified 11 persistent lineages local to Alberta, which were associated with 38.0% (95% CI 29.3%, 47.3%) of human isolates. During the later period, six locally persistent lineages continued

to be associated with human illness, including 74.7% (95% CI 68.3%, 80.3%) of reported cases in 2018 and 2019. Our study identified multiple locally evolving lineages transmitted between cattle and humans persistently associated with *E. coli* O157:H7 illnesses for up to 13 y. Locally persistent lineages may be a principal cause of the high incidence of *E. coli* O157:H7 in locations such as Alberta and provide opportunities for focused control efforts.

## Introduction

Several areas around the globe experience exceptionally high incidence of Shiga toxin-producing *Escherichia coli* (STEC), including the virulent serotype *E. coli* O157:H7. These include Scotland, (*Innocent et al., 2005*) Ireland, (*Óhaiseadha et al., 2017*) Argentina, (*Rivero et al., 2010*) and the Canadian province of Alberta (*Lisboa et al., 2019*). All are home to large populations of agricultural ruminants, STEC's primary reservoir. However, there are many regions with similar ruminant populations where STEC incidence is unremarkable. What differentiates high-risk regions is unclear. Moreover, with systematic STEC surveillance only conducted in limited parts of the world, (*Kirk et al., 2015*) there may be unidentified regions with exceptionally high disease burden.

STEC infections can arise from local reservoirs, transmitted through food, water, direct animal contact, or contact with contaminated environmental matrices. The most common reservoirs include domesticated ruminants such as cattle, sheep, and goats. Animal contact and consumption of contaminated meat and dairy products are significant risk factors for STEC, as are consumption of leafy greens, tomatoes, and herbs and recreational swimming that have been contaminated by feces from domestic ruminants (*FAO/WHO, 2018*) While STEC has been isolated from a variety of other animal species and outbreaks have been linked to species such as deer (*Laidler et al., 2013*) and swine, (*Honish et al., 2014*) it is unclear what roles they play as maintenance or intermediate hosts. STEC infections can be imported through food items traded nationally and internationally, as has been seen with *E. coli* O157:H7 outbreaks in romaine lettuce from the United States (*Bottichio et al., 2020*). Secondary transmission is believed to cause approximately 15% of cases, but transmission of the

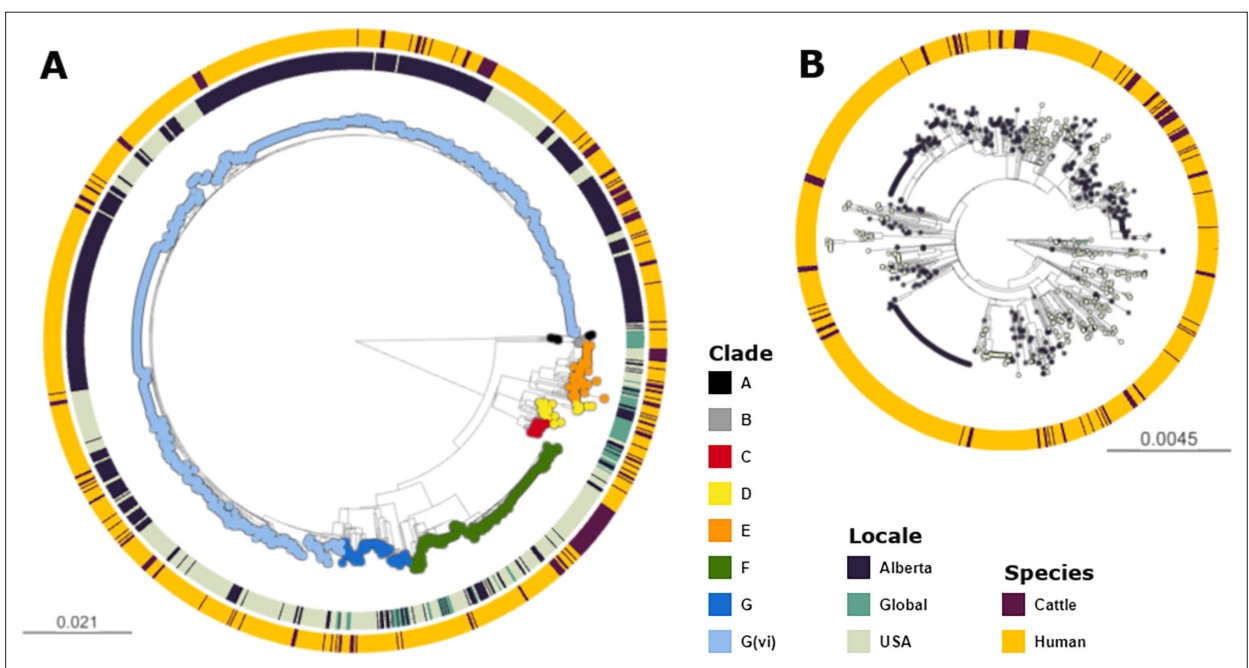

**Figure 1.** Maximum likelihood core SNP tree of the 1215 *E. coli* O157:H7 isolates referenced in the study. This includes 659 isolates from Alberta, Canada, from 2007 through 2019, 494 isolates from the U.S. from 1996 through 2019, and 62 isolates from elsewhere around the globe from 2007–2016. The tree was rooted at clade A. (**A**) Shows all clades, with tips colored according to clade, geographic origin shown on the inner ring, and species of origin on the outer ring. (**B**) shows clade G(vi), which constituted 73.6% of all isolates and 88.3% of Alberta isolates. Tips are colored by geographic origin, and the ring indicates species of origin.

**Table 1.** Distribution of study isolates by geographic source, clade, and Shiga toxin gene (*stx*) profile.

| Clade | Source | *stx1a* | *stx1a/stx2a* | *stx1a/stx2c* | *stx1a/stx2a/stx2c* | *stx2a* | *stx2c* | *stx2a/stx2c* | *stx2a/stx2c/stx2d* | *stx2a/stx2d* | None detected | Total |
|---|---|---|---|---|---|---|---|---|---|---|---|---|
| G(vi) | | 2 | 682 | 0 | 0 | 210 | 0 | 0 | 0 | 0 | 0 | 894 |
| | Alberta | 0 | 443 | 0 | 0 | 139 | 0 | 0 | 0 | 0 | 0 | 582 |
| | The U.S. | 2 | 237 | 0 | 0 | 71 | 0 | 0 | 0 | 0 | 0 | 310 |
| | Global | 0 | 2 | 0 | 0 | 0 | 0 | 0 | 0 | 0 | 0 | 2 |
| Other G | | 3 | 1 | 4 | 2 | 4 | 33 | 6 | 0 | 0 | 5 | 58 |
| | Alberta | 0 | 0 | 4 | 0 | 0 | 10 | 1 | 0 | 0 | 0 | 15 |
| | The U.S. | 2 | 1 | 0 | 2 | 1 | 19 | 4 | 0 | 0 | 3 | 32 |
| | Global | 1 | 0 | 0 | 0 | 3 | 4 | 1 | 0 | 0 | 2 | 11 |
| F | | 0 | 0 | 0 | 0 | 53 | 23 | 75 | 5 | 1 | 7 | 164 |
| | Alberta | 0 | 0 | 0 | 0 | 12 | 9 | 12 | 0 | 1 | 1 | 35 |
| | The U.S. | 0 | 0 | 0 | 0 | 33 | 11 | 61 | 5 | 0 | 6 | 116 |
| | Global | 0 | 0 | 0 | 0 | 8 | 3 | 2 | 0 | 0 | 0 | 13 |
| Other (A-E) | | 2 | 1 | 41 | 0 | 1 | 50 | 2 | 0 | 0 | 2 | 99 |
| | Alberta | 1 | 0 | 15 | 0 | 0 | 11 | 0 | 0 | 0 | 0 | 27 |
| | The U.S. | 0 | 0 | 10 | 0 | 1 | 24 | 1 | 0 | 0 | 0 | 36 |
| | Global | 1 | 1 | 16 | 0 | 0 | 15 | 1 | 0 | 0 | 2 | 36 |
| Total | | 7 | 684 | 45 | 2 | 268 | 106 | 83 | 5 | 1 | 14 | 1215 |

pathogen is not believed to be sustained through person-to-person transmission over the long term (*Tack et al., 2021*; *Kintz et al., 2017*).

The mix of STEC infection sources in a region directly influences public health measures needed to control disease burden. Living near cattle and other domesticated ruminants has been linked to STEC incidence, particularly for *E. coli* O157:H7 (*Óhaiseadha et al., 2017*; *Jaros et al., 2013*; *Elson et al., 2018*; *Frank et al., 2008*; *Friesema et al., 2011*; *Mulder et al., 2020*) These studies suggest an important role for local reservoirs in STEC epidemiology. A comprehensive understanding of STEC's disease ecology would enable more effective investigations into potential local transmission systems and ultimately their control. Here, we take a phylodynamic, genomic epidemiology approach to more precisely discern the role of the cattle reservoir in the dynamics of *E. coli* O157:H7 human infections. We focus on the high incidence region of Alberta, Canada to provide insight into characteristics that make the pathogen particularly prominent in such regions.

## Results
### Description of isolates
Across the 1215 isolates included in the analyses, we identified 12,273 core genome SNPs. Clade G(vi) constituted 73.6% (n=894) of all isolates (*Figure 1a*). Clade A, which is the most distinct of the *E. coli* O157:H7 clades, included non-Alberta isolates, two human isolates from Alberta, and no Alberta cattle isolates. The majority of all Alberta isolates belonged to the G(vi) clade (582 of 659; 88.3%), compared to 281 of the 1560 (18.0%) randomly sampled U.S. PulseNet isolates that were successfully assembled and QCed. Among the 62 non-randomly sampled global isolates, only 2 (3.2%) were clade G(vi) (*Figure 1b*). There were 682 (76.3%) clade G(vi) isolates with the *stx1a/stx2a* profile and 210

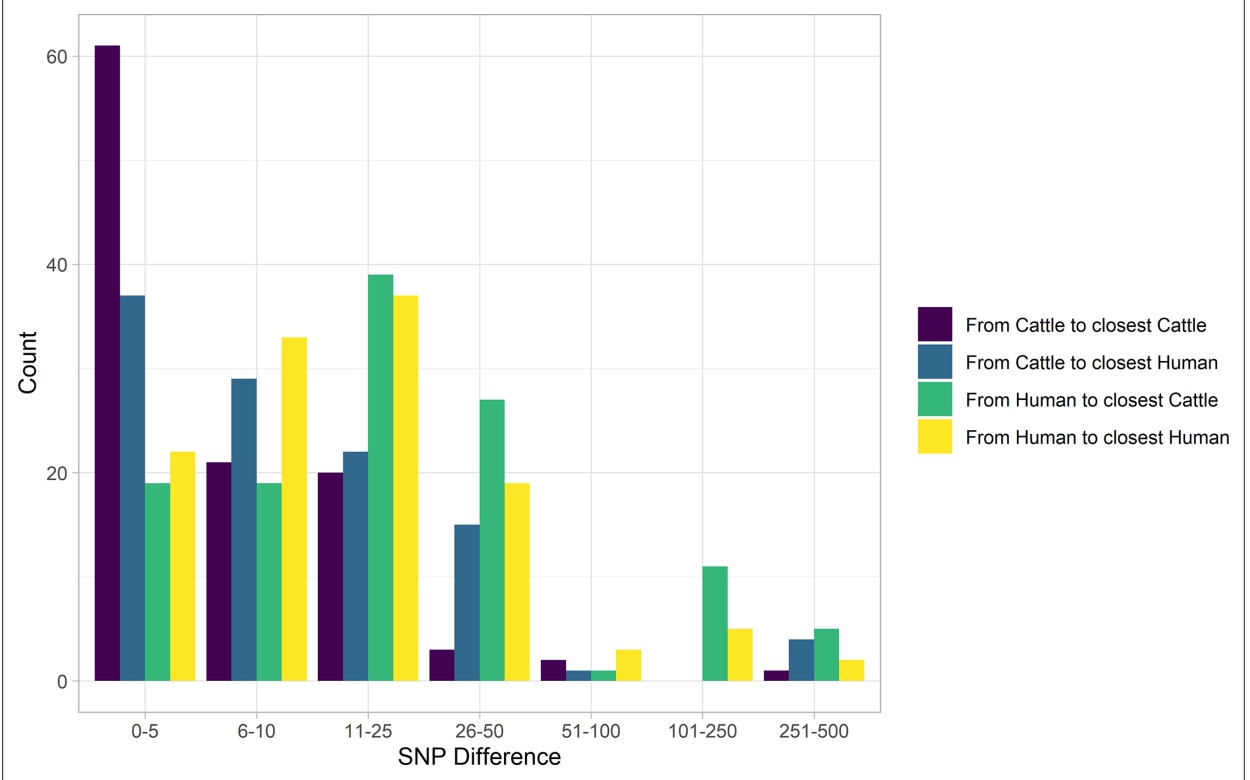

**Figure 2.** SNP distance between *E. coli* O157:H7 strains and their nearest relative, by species, Alberta, Canada, 2007–2015. Distances for isolates from 121 reported human cases and 108 beef cattle are shown. Cattle isolates were highly related with 56.5% of cattle isolates within five SNPs of another cattle isolate and 94.4% within 25 SNPs. Human isolates showed a bimodal distribution in their relationship to cattle isolates, with 86.0% within 50 SNPs of a cattle isolate and the remainder 185–396 SNPs apart. Nineteen human isolates (15.7%) were within five SNPs of a cattle isolate.

(23.5%) with the *stx2a*-only profile, compared to 2 (0.6%) and 58 (18.1%), respectively, among the 321 isolates outside the G(vi) clade (*Table 1*).

## The majority of clinical cases evolved from local cattle lineages

In our primary sample of 121 human and 108 cattle isolates from Alberta from 2007 to 2015, SNP distances were comparable between species (*Figure 2*). Among sampled human cases, 19 (15.7%; 95% CI 9.7%, 23.4%) were within five SNPs of a sampled cattle strain. The median SNP distance between cattle sequences was 45 (IQR 36–56), compared to 54 (IQR 43–229) SNPs between human sequences from cases in Alberta during the same years.

The phylogeny generated by our primary structured coalescent analysis indicated cattle were the primary reservoir, with a high probability that the hosts at nodes along the backbone of the tree were cattle (*Figure 3*). The root was estimated at 1802 (95% HPD 1731, 1861). The most recent common ancestor (MRCA) of clade G(vi) strains in Alberta was inferred to be a cattle strain, dated to 1969 (95% HPD 1959, 1979). With our assumption of a relaxed molecular clock, the mean clock rate for the core genome was estimated at $9.65×10^{-5}$ (95% HPD $8.13×10^{-5}$, $1.13×10^{-4}$) substitutions/site/year. The effective population size, $N_e$, of the human *E. coli* O157:H7 population, was estimated as 1060 (95% HPD 698, 1477), and for cattle as 73 (95% HPD 50, 98). We estimated 108 (95% HPD 104, 112) human lineages arose from cattle lineages, and 14 (95% HPD 5, 23) arose from other human lineages (*Figure 3*). In other words, 88.5% of human lineages seen in Alberta from 2007 to 2015 arose from cattle lineages. We observed minimal influence of our choice of priors (*Figure 3—figure supplement 1*). Our sensitivity analysis of equal numbers of isolates from cattle and humans was largely consistent with our primary results, estimating that 94.3% of human lineages arose from cattle lineages (*Figure 3—figure supplement 2*).

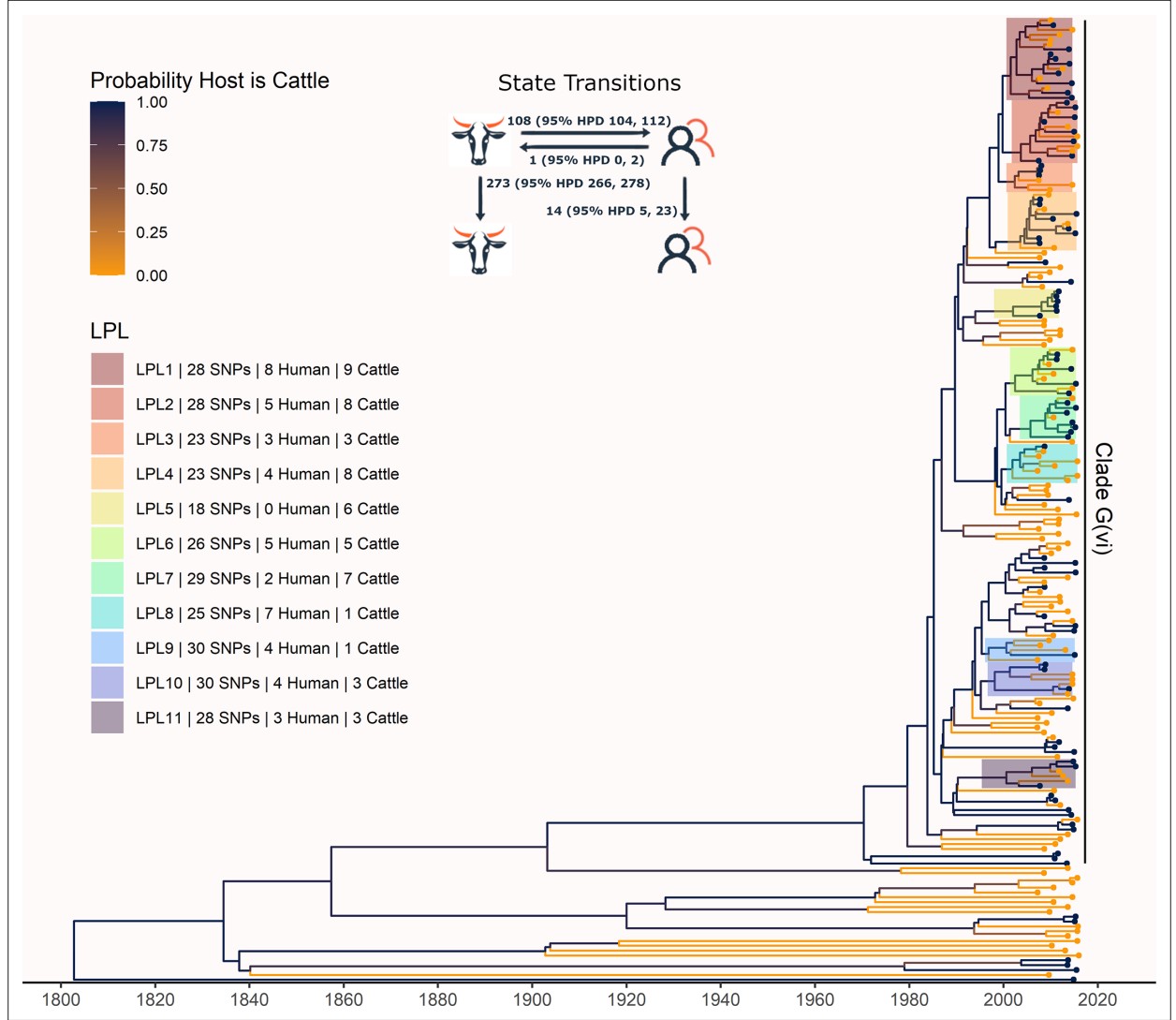

**Figure 3.** Maximum clade credibility (MCC) tree of structured coalescent analysis of *E. coli* O157:H7 strains isolated from 115 reported human cases and 84 beef cattle in Alberta, Canada, 2007–2015. Isolates were down-sampled prior to phylodynamic analysis to remove isolates that were highly similar. The structured coalescent analysis estimated migration and state transitions between humans and cattle. The MCC tree was colored by inferred host, cattle (blue) or human (orange). The minimum SNP distance between all isolates in the LPL, and the number of human and cattle isolates after down-sampling are shown for each LPL. The majority of ancestral nodes inferred as cattle suggest cattle as the primary reservoir. The root was estimated at 1802 (95% HPD 1731, 1861). Eleven locally persistent lineages (LPLs) were identified, all in the G(vi) clade and labeled LPL 1 through 11. For each LPL, the minimum number of SNPs all isolates are within (30 was set as the maximum) and the number of human and cattle isolates within the LPL, not including down-sampled isolates, are shown. With down-sampled isolates reincorporated, LPLs accounted for 46 human (38.0%) and 71 cattle (65.7%) isolates. The structured coalescent model estimated 108 cattle-to-human state transitions between branches, compared to only 14 human-to-human transitions, inferring cattle as the origin of 88.5% of human lineages.

The online version of this article includes the following figure supplement(s) for figure 3:

**Figure supplement 1.** Key parameters are drawn from the posterior distributions of four Markov chain Monte Carlo chains using different starting seeds compared to draws from the prior distribution.

**Figure supplement 2.** Maximum clade credibility (MCC) tree of structured coalescent analysis of 168 subsampled isolates from humans and cattle from Alberta, 2007–2015.

**Figure supplement 3.** Sampled trees from four independent chains of the Alberta 2007–2015 structured coalescent analysis.

**Figure supplement 4.** LPLs from the primary analysis were defined using alternate SNP thresholds of 50 (left) and 75 (right) SNPs.

## Locally persistent lineages account for the majority of ongoing human disease

In our primary analysis, we identified 11 locally persistent lineages (LPLs) (*Figure 3*). After reincorporating down-sampled isolates, LPLs included a range of 5 (G(vi)-AB LPL 9)–26 isolates (G(vi)-AB LPL 1), with an average of 10. LPL assignment was based on the MCC tree of the combination of four independent chains. LPLs persisted for 5–9 y, with the average LPL spanning 8 y. By definition, MRCAs of each LPL were required to have a posterior probability ≥95% on the MCC tree, and in practice, all had posterior probabilities of 99.7–100%. Additionally, examining all trees sampled from the four chains supported the same major lineages (*Figure 3—figure supplement 3*). Our sensitivity analysis of equal numbers of isolates from cattle and humans identified 10 of the same 11 LPLs (*Figure 3—figure supplement 2*). G(vi)-AB LPL 9 was no longer identified as an LPL, because it fell below the five-isolate threshold after subsampling. Additionally, G(vi)-AB LPL 8 expanded to include a neighboring branch.

LPLs tended to be clustered on the MCC tree. G(vi)-AB LPLs 1–4, 6–8, and 9 and 10 were clustered with MRCAs inferred in 1996 (95% HPD 1992, 1999), 1998 (95% HPD 1995, 2000), and 1993 (95% HPD 1989, 1996), respectively (*Figure 3*). Cattle were the inferred host of all three ancestral nodes. LPLs were assigned using a threshold of 30 SNPs. In sensitivity analysis testing alternate SNP thresholds, we observed LPLs mimicking the larger clusters of the LPLs from our primary analysis (*Figure 3—figure supplement 4*).

LPLs included 71 of 108 (65.7%; 95% CI 56.0%, 74.6%) cattle and 46 of 121 (38.0%; 95% CI 29.3%, 47.3%) human isolates. Of the remaining human isolates, 33 (27.3%) were associated with imported infections and 42 (34.7%) with infections from transient local strains. Of the remaining cattle isolates, 11 (10.2%) were imported and 26 (24.1%) were associated with transmission from transient strains. Of the 117 isolates in LPLs, 7 (6.0%) carried only *stx2a*, and the rest *stx1a/stx2a*. Among the 112 non-LPL isolates, 1 (0.9%) was *stx1a*-only, 27 (24.1%) were *stx2a*-only, 5 (4.5%) were *stx2c*-only, 68 (60.7%) were *stx1a/stx2a*, 6 (5.4%) were *stx1a/stx2c*, and 5 (4.5%) were *stx2a/stx2c*.

To understand long-term persistence, we expanded the phylogeny with additional Alberta Health isolates from 2009 to 2019. Six of the 11 LPLs identified in our primary analysis, G(vi)-AB LPLs 1, 2, 4, 7, 10, and 11, continued to cause disease during the 2016–2019 period (*Figure 4*). With most cases reported during 2018 and 2019 sequenced, we were able to estimate the proportion of reported *E. coli* O157:H7 associated with LPLs. Of 217 sequenced cases reported during these 2 y, 162 (74.7%; 95% CI 68.3%, 80.3%) arose from Alberta LPLs. The *stx* profile of LPL isolates shifted as compared to the primary analysis, with 83 (51.2%; 95% CI 43.3%, 59.2%) of the LPL isolates encoding only *stx2a* and the rest *stx1a/stx2a* (*Figure 5*). Among the 55 non-LPL isolates during 2018–2019, the *stx2c*-only profile emerged with 16 (29.1%; 95% CI 17.6%, 42.9%) isolates, *stx2a*-only was found in six (10.9%; 95% CI 4.1%, 22.2%) isolates, and five (9.1%; 95% CI 3.0%, 20.0%) isolates carried both *stx2a* and *stx2c*.

All five large (≥10 cases) sequenced outbreaks in Alberta during the study period were within clade G(vi). G(vi)-AB LPLs 2 and 7 gave rise to three large outbreaks, accounting for 117 cases (both sequenced and unsequenced), including 83 from an extended outbreak by a single strain in 2018 and 2019, defined as isolates within five SNPs of one another. The two large outbreaks that did not arise from LPLs both occurred in 2014 and were responsible for 164 cases.

## Locally persistent lineages were not imported

Of the 494 U.S. isolates analyzed, nine (1.8%; 95% CI 0.8%, 3.4%) occurred within Alberta LPLs after re-incorporating down-sampled isolates (*Figure 6*). None of the 62 global isolates were associated with Alberta LPLs. The 9 U.S. isolates were part of G(vi)-AB LPLs 2 (n=3), 4 (n=4), 7 (n=1), and 11 (n=1), all of which had Alberta isolates that spanned 9–13 y and predated the U.S. isolates. There was no evidence of U.S. or global ancestors of LPLs. Based on migration events calculated from the structured tree, we estimated that 11.0% of combined human and cattle Alberta lineages were imported (*Table 2*). Alberta sequences were separated from U.S. and global sequences by a median of 63 (IQR 45–236) and 225 (IQR 209–249) SNPs, respectively.

Including U.S. and global isolates in the phylogeny did not change which LPLs we identified (*Figure 6*). The minimum SNPs that LPL isolates differed by was lower than in the Alberta-only analyses, because the core genome shared by all Alberta, U.S., and global isolates was smaller than that of only the Alberta isolates. Alberta sequences included in some LPLs changed slightly. G(vi)-AB LPL

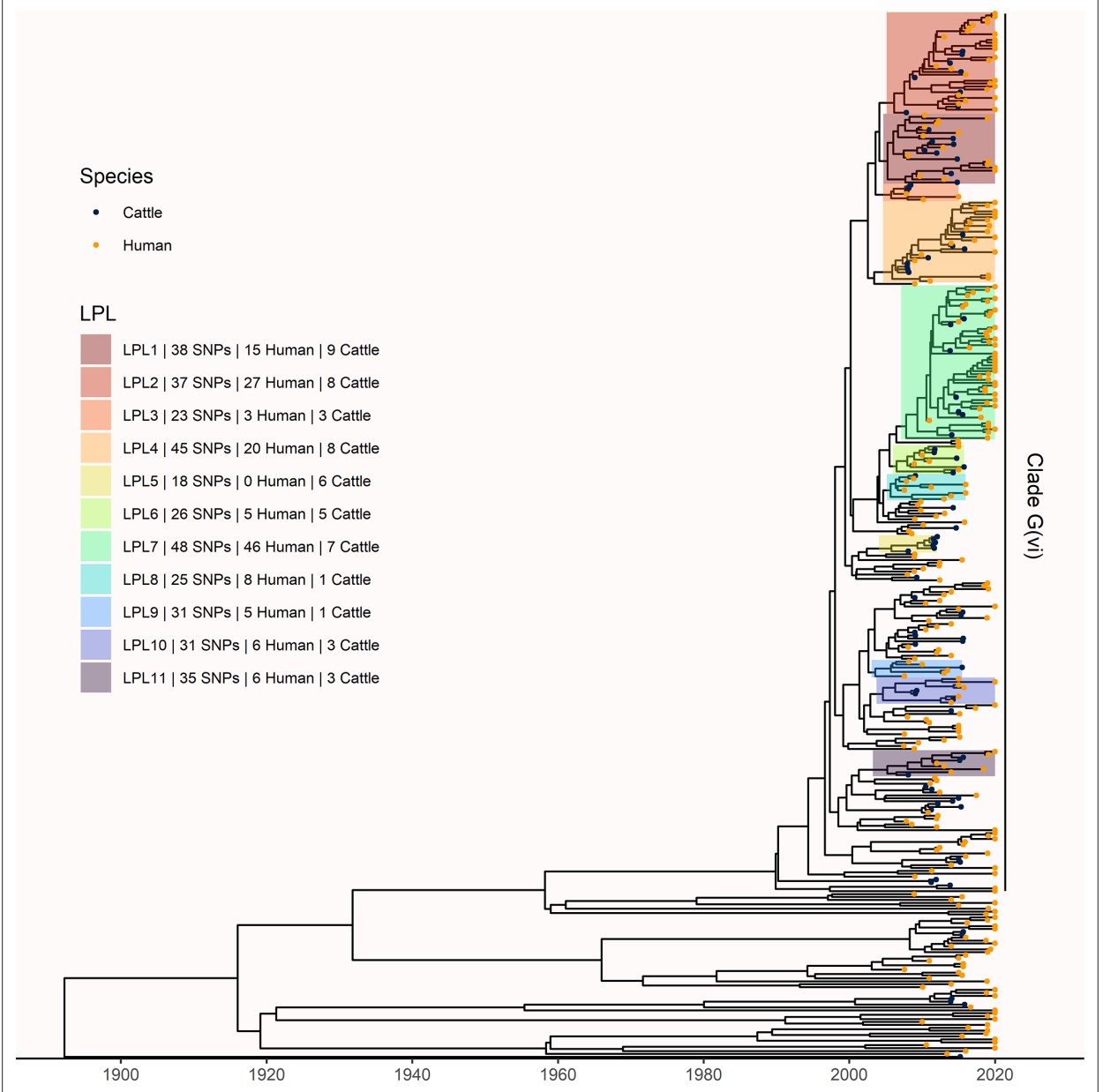

**Figure 4.** Extension of Alberta, Canada *E. coli* O157:H7 analysis to include 229 randomly selected study isolates and 430 additional public health isolates available from 2009–2019. The maximum clade credibility (MCC) tree was constructed from a coalescent analysis with constant population size after down-sampling. Six locally persistent lineages (LPLs) in clade G(vi) continued to be associated with the disease after the initial study period. LPLs are colored and labeled as in *Figure 3*. After re-incorporating the down-sampled isolates, 74.7% of reported cases in 2018 and 2019 were associated with an LPL.

4 lost three Alberta isolates from clinical cases, and G(vi)-AB LPLs 6 and 11 both lost one cattle and one human isolate from Alberta. In these LPLs, the isolates no longer included were the most outlying isolates in the LPLs defined using only Alberta isolates (*Figure 4*). Of the 217 Alberta human isolates from 2018 and 2019, 160 (73.7%) were still associated with LPLs after the addition of U.S. and global isolates, demonstrating the stability of the extended analysis results.

## Discussion

Focusing on a region that experiences an especially high incidence of STEC, we conducted a deep genomic epidemiologic analysis of *E. coli* O157:H7's multi-host disease dynamics. Our study identified

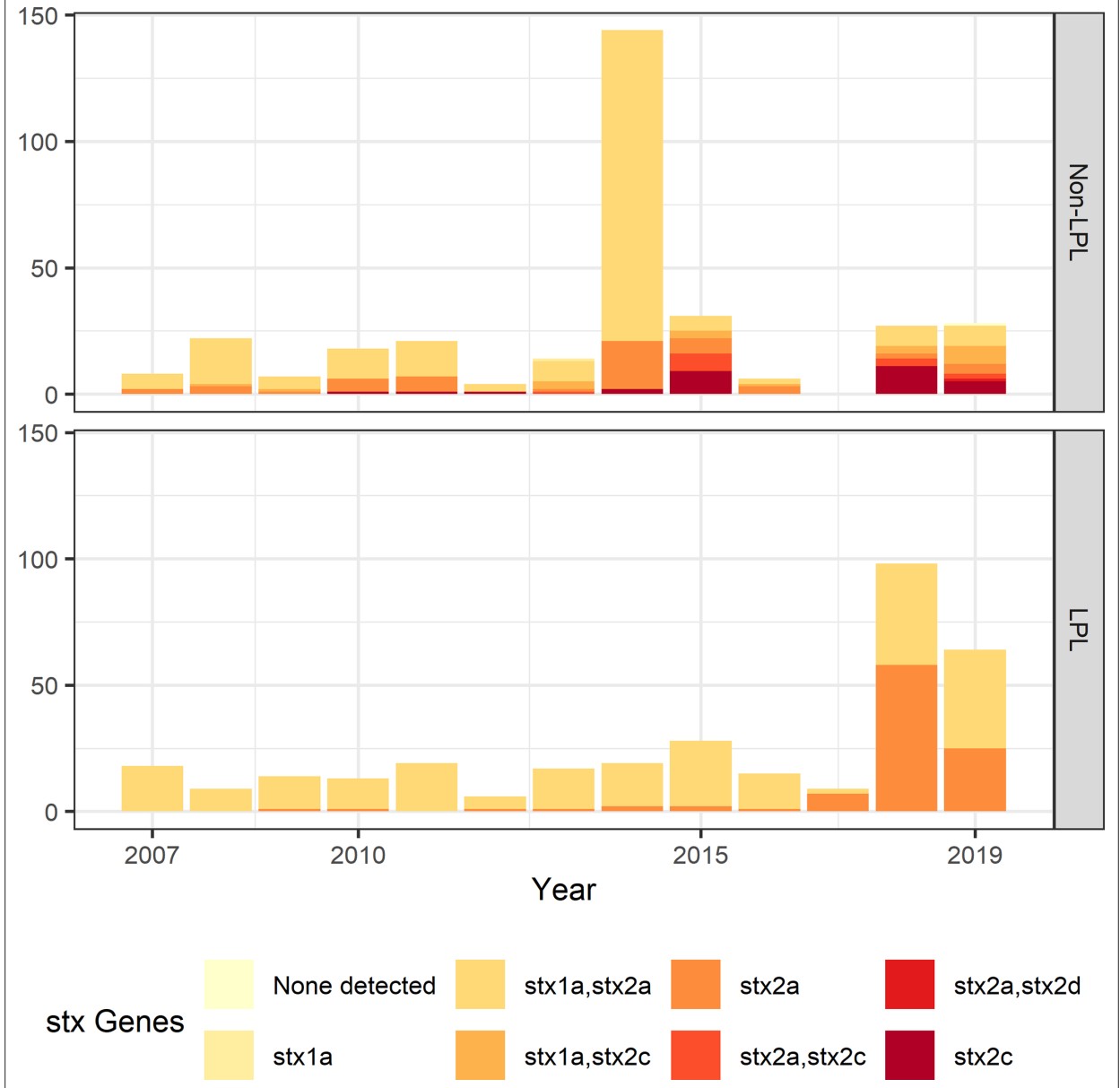

**Figure 5.** Shiga toxin gene (*stx*) profile by locally persistent lineage (LPL) status of extended analysis of Alberta, Canada *E. coli* O157:H7 isolated from cattle and humans, 2007–2019. The *stx* profile across all clades shifted from the initial study period (2007–2015) to the later study period (2016–2019), with more of the virulent *stx2a*-only profile observed in 2018 and 2019 than in previous years. In 2018 and 2019, 51.2% of LPL isolates carried only *stx2a*, compared to 10.9% of non-LPL isolates. The peak in sequences in 2014 is due to two outbreaks; routine sequencing began in 2018 and 2019, accounting for the rise in sequenced cases during those years.

multiple locally evolving lineages transmitted between cattle and humans. These were persistently associated with *E. coli* O157:H7 illnesses over periods of up to 13 y, the length of our study. Of clinical importance, there was a dramatic shift in the *stx* profile of the strains arising from locally persistent lineages toward strains carrying only *stx2a*, which has been associated with increased progression to hemolytic uremic syndrome (HUS) (*Tarr et al., 2019*).

Our study has provided quantitative estimates of cattle-to-human migration in a high-incidence region, the first such estimates of which we are aware. Our estimates are consistent with prior work that established an increased risk of STEC associated with living near cattle (*Óhaiseadha et al., 2017*; *Jaros et al., 2013*; *Elson et al., 2018*; *Frank et al., 2008*; *Friesema et al., 2011*; *Mulder et al., 2020*). We showed that 88.5% of strains infecting humans arose from cattle lineages. These transitions

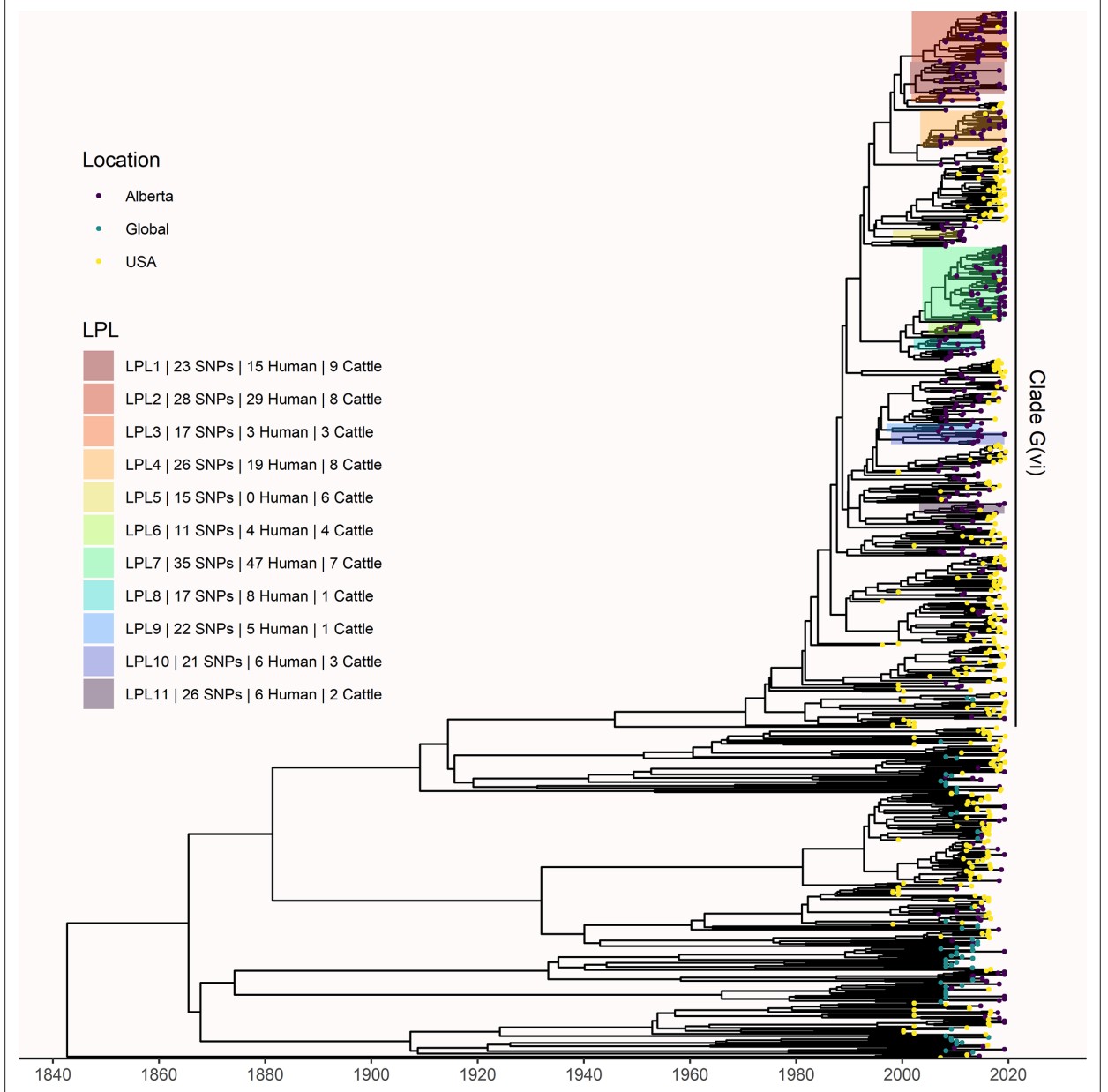

**Figure 6.** Comparison of Alberta, U.S., and global *E. coli* O157:H7 isolates. Tips are colored based on isolate origin, and locally persistent lineages (LPLs) from *Figure 3* are highlighted. A total of nine U.S. isolates arose from Alberta LPLs 2, 4, 7, and 11, all of which had Alberta isolates predating the U.S. isolates. No global isolates were associated with Alberta LPLs. Clade A was excluded from the analysis due to its high level of divergence from the rest of the *E. coli* O157:H7 population.

can be seen as a combination of the infection of humans from local cattle or cattle-related reservoirs in clade G(vi) and the historic evolution of *E. coli* O157:H7 from cattle in the rare clades. While our findings indicate the majority of human cases arose from cattle lineages, transmission may involve intermediate hosts or environmental sources several steps removed from the cattle reservoir. Small ruminants (e.g. sheep, goats) have been identified as important STEC reservoirs, (*Elson et al., 2018*; *Mulder et al., 2020*; *Strachan et al., 2015*) and Alberta has experienced outbreaks linked to swine (*Honish et al., 2014*). Exchange of strains between cattle and other animals may occur if co-located, if surface water sources near farms are contaminated, and through wildlife, including deer, birds, and flies (*Ladd-Wilson et al., 2022*; *Rapp et al., 2021*) Humans can also become infected from environmental sources, such as through swimming in contaminated water. Although transmission systems may be multi-faceted, our analysis demonstrates that local cattle remain an integral part of

**Table 2.** Estimated migrations from structured coalescent analysis of Alberta, U.S., and global isolates, excluding clade A.

| Migration direction | Mean | 95% HPD |
|---|---|---|
| Alberta to Alberta | 589 | 570, 609 |
| Alberta to Global | 8 | 4, 11 |
| Alberta to U.S. | 12 | 6, 16 |
| Global to Alberta | 68 | 57, 78 |
| Global to Global | 442 | 363, 511 |
| Global to U.S. | 296 | 267, 335 |
| U.S. to Alberta | 5 | 0, 14 |
| U.S. to Global | 13 | 0, 35 |
| U.S. to U.S. | 104 | 10, 184 |

HPD, highest posterior density interval.

the transmission system for the vast majority of cases, even when they may not be the immediate source of infection. Indeed, despite our small sample of *E. coli* O157:H7 isolates from cattle, 15.7% of our human cases were within five SNPs of a cattle isolate, suggesting that cattle were a recent source of transmission, either through direct contact with the animal or their environments or consumption of contaminated food products.

The cattle-human transitions we estimated were based on structured coalescent theory, which we used throughout our analyses. This approach is similar to other phylogeographic methods that have previously been applied to *E. coli* O157:H7 (*Franz et al., 2019*). We inferred the full backbone of the Alberta *E. coli* O157:H7 phylogeny as arising from cattle, consistent with the postulated global spread of the pathogen via ruminants (*Franz et al., 2019*). Our estimate of the origin of the serotype, at 1802 (95% HPD 1731, 1861), was somewhat earlier than previous estimates, but consistent with global (1890; 95% HPD 1845, 1925) *Franz et al., 2019* and the United Kingdom (1840; 95% HPD 1817, 1855) (*Dallman et al., 2015*) studies that used comparable methods. Our dating of the G(vi) clade in Alberta to 1969 (95% HPD 1959, 1979) also corresponds to proposed migrations of clade G into Canada from the U.S. in 1965–1977 (*Franz et al., 2019*). Our study thus adds to the growing body of work on the larger history of *E. coli* O157:H7, providing an in-depth examination of the G(vi) clade.

Our identification of the 11 locally persistent lineages (LPLs) is significant in demonstrating that the majority of Alberta's reported *E. coli* O157:H7 illnesses are of local origin. Our definition ensured that every LPL had an Alberta cattle strain and at least five isolates separated by at least 1 y, making the importation of the isolates in a lineage highly unlikely. For an LPL to be fully imported, cattle and human isolates would need to be repeatedly imported from a non-Alberta reservoir where the lineage persisted over several years. Further supporting the evolution of the LPLs within Alberta, all 11 LPLs were in clade G(vi), several were phylogenetically related with MRCAs dating to the late 1990 s, and few non-Alberta isolates fell within LPLs. The nineU.S. isolates associated with Alberta LPLs may reflect Alberta cattle that were slaughtered in the U.S. or infections in travelers from the U.S. Thus, we are confident that the identified LPLs represent locally evolving lineages and potential persistent sources of disease. We also showed that the identification of these LPLs was robust to the sampling strategy, with only the smallest LPL failing to be identified after subsampling left it with <5 isolates.

We estimated the proportion of *E. coli* O157:H7 that were imported into Alberta in two ways. Based on our LPL analysis, we estimated only 27% of human and 10% of cattle *E. coli* O157:H7 isolates were imported. This was slightly higher than the overall importation estimate of 11% for all Alberta lineages from our global structured coalescent analysis. Our global structured coalescent analysis also estimated that 3% of lineages in the U.S. and 2% of lineages outside the U.S. and Canada had been exported from Alberta, suggesting that Alberta is not a significant contributor to the global *E. coli* O157:H7 burden beyond its borders. These results place the *E. coli* O157:H7 population in Alberta within a larger context, indicating that the majority of diseases can be considered local. At least one study has attempted to differentiate local vs. non-local lineages based on travel status, (*Dallman et al., 2022*) of which may be appropriate in some locations but can miss cases imported through food

products, such as produce imported from other countries. To our knowledge, our study provides the first comprehensive determination of local vs. imported status for *E. coli* O157:H7 cases using external reference cases. Similar studies in regions of both high and moderate incidence would provide further insight into the role of localization on *E. coli* O157:H7 incidence.

Of the 11 lineages we identified as LPLs during the 2007–2015 period, six were also associated with cases that occurred during the 2016–2019 period. During the initial period, 38% of human cases were linked to an LPL, and 6% carried only *stx2a*. The risk of HUS increases in strains of STEC carrying only *stx2a*, relative to *stx1a/stx2a*, (*Tarr et al., 2019*) meaning the earlier LPL population had fewer high-virulence strains. In 2018 and 2019, the six long-term LPLs were associated with both greater incidence and greater virulence, encompassing 75% of human cases with more than half of LPL isolates carrying only *stx2a*. The cause of this shift remains unclear, though shifts toward greater virulence in *E. coli* O157:H7 populations have been seen elsewhere (*Byrne et al., 2018*). The growth and diversity of G(vi)-AB LPLs 2, 4, and 7 in the later period suggest these lineages were in stable reservoirs or adapted easily to new niches. Identifying these reservoirs could yield substantial insights into the disease ecology that supports LPL maintenance and opportunities for disease prevention, given the significant portion of illnesses caused by persistent strains.

The high proportion of cases associated with cattle-linked local lineages is consistent with what is known about the role of cattle in STEC transmission. Among sporadic STEC infections, 26% have been attributed to animal contact and the farm environment, with a further 19% to pink or raw meat (*Kintz et al., 2017*). Similarly, 24% of *E. coli* O157 outbreaks in the U.S. have been attributed to beef, animal contact, water, or other environmental sources (*Tack et al., 2021*). In Alberta, these are all inherently local exposures, given that 90% of beef consumed in Alberta is produced and/or processed there. Even person-to-person transmission, responsible for 15% of sporadic cases and 16% of outbreaks, (*Tack et al., 2021*; *Kintz et al., 2017*) includes secondary transmission from cases infected from local sources, which may explain our estimate of 11.5% of human lineages arising from other human lineages.

We developed a novel measure of persistence for use in this study, specifically for the purposes of identifying lineages that pose an ongoing threat to public health in a specific region. Persistence has been variably defined in the literature, for example, as shedding of the same strain for at least 4 mo (*Barth et al., 2016*). Most recently, the U.S. CDC identified the first Recurring, Emergent, and Persistent (REP) STEC strain, REPEXH01, an *E. coli* O157:H7 strain detected since 2017 in over 600 cases. REPEXH01 strains are within 21 allele differences of one another (https://www.cdc.gov/ncezid/dfwed/outbreak-response/rep-strains/repexh01.html), and REP strains from similar enteric pathogens are defined based on allele differences of 13–104. Given that we used high-resolution SNP analysis rather than cgMLST, we used a difference of ≤30 SNPs to define persistent lineages. While both our study and the REPEXH01 strain identified by the CDC indicate that persistent strains of *E. coli* O157:H7 exist, the O157:H7 serotype was defined as sporadic in a German study using the 4 mo shedding definition (*Barth et al., 2016*). This may be due to strain differences between the two locations, but it might also indicate that persistence occurs at the host community level, rather than the individual host level. Understanding microbial drivers of persistence is an active field of research, with early findings suggesting a correlation of STEC persistence to the accessory genome and traits such as biofilm formation and nutrient metabolism (*Barth et al., 2016*; *Barth et al., 2020*). Our approach to studying persistence was specifically designed for longitudinal sampling in high-incidence regions and may be useful for others attempting to identify sources that disproportionately contribute to disease burden. Although we used data from the reservoir species to help define the LPLs in this study, we are testing alternate approaches that rely on only routinely collected public health data.

We limited our analysis to *E. coli* O157:H7 despite the growing importance of non-O157 STEC, as historical multi-species collections of non-O157 isolates are lacking. As serogroups differ meaningfully in exposures, (*Tarr et al., 2023*) our results may not be generalizable beyond the O157 serogroup. However, cattle are still believed to be a primary reservoir for non-O157 STEC, and cattle density is associated with the risk of several non-O157 serogroups (*Frank et al., 2008*). Person-to-person transmission remains a minor contributor to the STEC burden. For all of these reasons, if we were to conduct this analysis in non-O157 STEC, we expect the majority of human lineages would arise from cattle lineages. Additionally, persistence within the cattle reservoir has been observed for a range of serogroups, (*Barth et al., 2016*) suggesting that LPLs also likely exist among non-O157 STEC.

Our findings may have implications beyond STEC, as well. Other zoonotic enteric pathogens such as *Salmonella* and *Campylobacter* can persist, and outbreaks are regularly linked to localized animal populations and produce-growing operations contaminated by animal feces. The U.S. CDC has also defined REP strains for these pathogens. LPLs could shed light on how and where persistent strains are proliferating, and thus where they can be controlled.

The identification of LPLs serves multiple purposes, because they suggest the existence of local reservoir communities that maintain specific strains for long periods. First, they further our understanding of the complex systems that allow STEC to persist. In this study, the LPLs we identified persisted for 5–13 y. The reservoir communities that enable persistence could involve other domestic and wild animals previously found to carry STEC (*Elson et al., 2018*; *Mulder et al., 2020*; *Strachan et al., 2015*; *Ladd-Wilson et al., 2022*; *Rapp et al., 2021*; *Szczerba-Turek et al., 2023*). The feedlot environment also likely plays an important role in persistence, as water troughs and pen floors have been identified as important sources of STEC for cattle (*Ayscue et al., 2009*). Identifying LPLs is a first step in identifying these reservoir communities and determining what factors enable strains to persist, so as to identify them for targeted interventions. Second, the identification of these LPLs in cattle could identify the specific local reservoirs of STEC. Similar to source tracing in response to outbreaks, LPLs provide an opportunity for cattle growers to identify cattle carrying the specific strains that are associated with a large share of human disease in Alberta. While routinely vaccinating against STEC has not been shown to be efficacious or cost-effective, (*Smith, 2014*) a ring-type vaccination strategy in response to an identified LPL isolate could overcome the limitations of previous vaccination strategies. Third, the identification of new clinical cases infected with LPL strains could help direct public health investigations toward local sources of transmission. Finally, the disease burden associated with LPLs could be compared across locations and may help explain how high-incidence regions differ from regions with lower incidence.

Our analysis was limited to only cattle and humans. Had isolates from a wider range of potential reservoirs been available, we would have been able to elucidate more clearly the roles that various hosts and common sources of infection play in local transmission. Additional hosts may help explain the 1 human-to-cattle predicted transmission, which could be erroneous. As with all studies utilizing public health data, sampling from only severe cases of disease is biased toward clinical isolates. In theory, this could limit the genetic variation among human isolates if virulence is associated with specific lineages. However, clinical isolates were more variable than cattle isolates, dominating the most divergent clade A, so the overrepresentation of severe cases does not appear to have appreciably biased the current study. Similarly, in initially selecting an equal number of human and cattle isolates, we sampled a larger proportion of the human-infecting *E. coli* O157:H7 population compared to the population that colonizes cattle. As cattle are the primary reservoir of *E. coli* O157:H7, the pathogen is more prevalent in cattle than in humans, who appear to play a limited role in sustained transmission. In sampling a larger proportion of the strains that infect humans, we likely sampled a wider diversity of these strains compared to those in cattle, which could have biased the analysis toward finding humans as the ancestral host. Thus, the proportion of human lineages arising from cattle lineages (88.5%) might be underestimated, which is also suggested by our sensitivity analysis of equal numbers of cattle and clinical isolates. Finally, we were not able to estimate the impact of strain migration between Alberta and the rest of Canada, because locational metadata for publicly available *E. coli* O157:H7 sequences from Canada was limited.

*E. coli* O157:H7 infections are a pressing public health problem in many high-incidence regions around the world including Alberta, where a recent childcare outbreak caused >300 illnesses. In the majority of sporadic cases, and even many outbreaks, (*Tack et al., 2021*) the source of infection is unknown, making it critical to understand the disease ecology of *E. coli* O157:H7 at a system level. Here, we have identified a high proportion of human cases arising from cattle lineages and a low proportion of imported cases. Local transmission systems, including intermediate hosts and environmental reservoirs, need to be elucidated to develop management strategies that reduce the risk of STEC infection. In Alberta, local transmission is dominated by a single clade, and over the extended study period, persistent lineages caused an increasing proportion of disease. The local lineages with long-term persistence are of particular concern because of their increasing virulence, yet they also present opportunities as larger, more stable targets for reservoir identification and control.

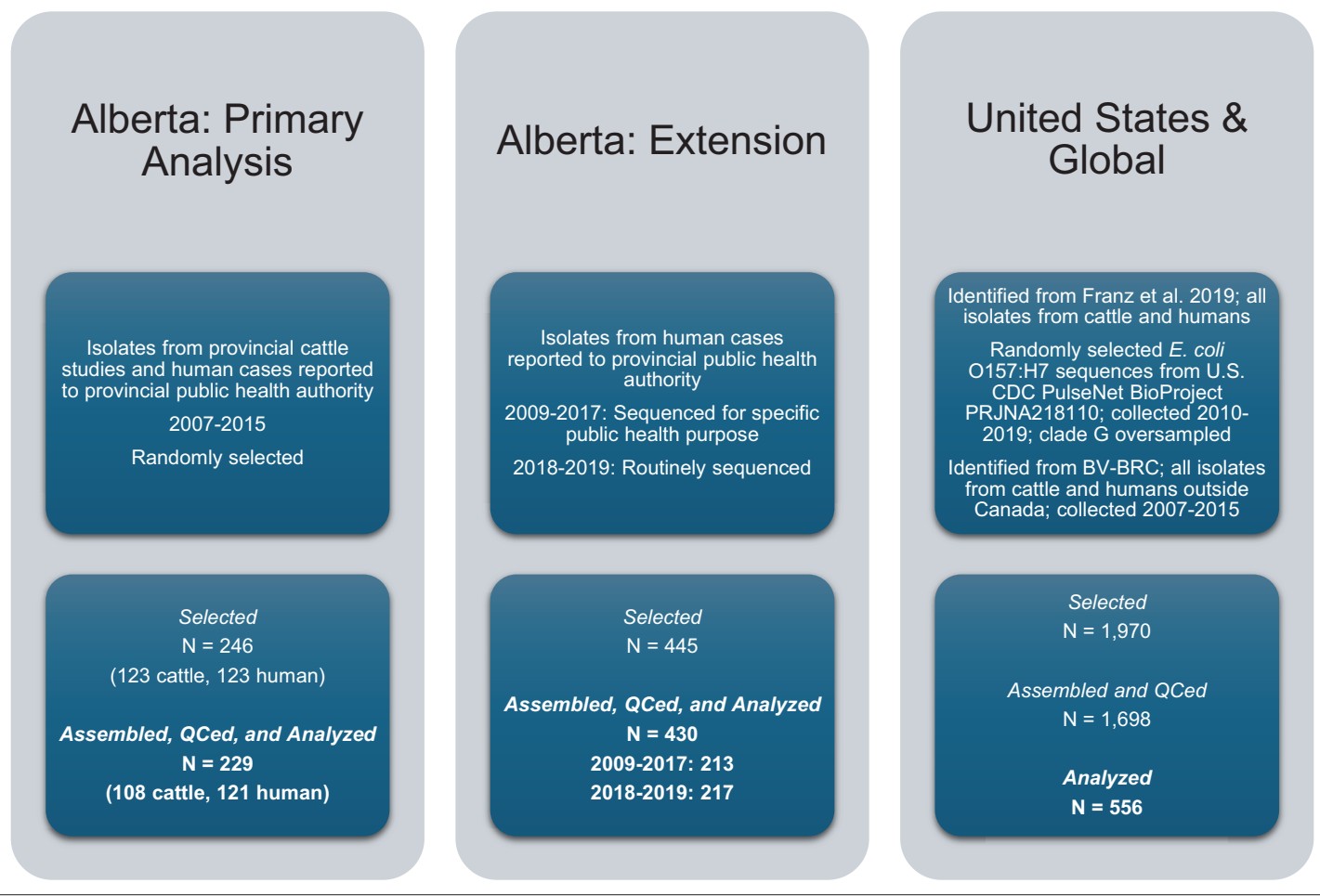

**Figure 7.** *E. coli* O157:H7 isolates were selected for the study and assembled. Three sets of isolates, all originating from cattle or humans, were included in the study.

The online version of this article includes the following figure supplement(s) for figure 7:

**Figure supplement 1.** Clustering was performed from raw reads using PopPUNK v2.5.0 with 10,146 *E.coli* reference genomes *Lees et al., 2019*.

## Materials and methods
### Study design and population

We conducted a multi-host genomic epidemiology study in Alberta, Canada. Our primary analysis focused on 2007–2015 due to the availability of isolates from intensive provincial cattle studies (*Stephens et al., 2009*; *Stanford et al., 2014*; *Stanford et al., 2013*; *Stanford et al., 2016*). These studies rectally sampled feces from individual animals, hide swabs, fecal pats from the floors of pens of commercial feedlot cattle, or feces from the floors of transport trailers. In studies of pens of cattle, samples were collected from the same cattle at least twice over a 4 to 6 mo period. A one-time composite sample was collected from cattle in transport trailers, which originated from feedlots or auction markets in Alberta. To select both cattle and human isolates, we block randomized by year to ensure representation across the period. We define isolates as single bacterial species obtained from culture. We sampled 123 *E. coli* O157 cattle isolates from 4660 available. Selected cattle isolates represented 7 of 12 cattle study sites and 56 of 89 sampling occasions from the source studies (*Stephens et al., 2009*; *Stanford et al., 2014*; *Stanford et al., 2013*; *Stanford et al., 2016*). We sampled 123 of 1148 *E. coli* O157 isolates collected from cases reported to the provincial health authority (Alberta Health) during the corresponding time period (Appendix 1).

In addition to the 246 isolates for the primary analysis, we contextualized our findings with two additional sets of *E. coli* O157:H7 isolates (*Figure 7*): 445 from Alberta Health from 2009 to 2019

and already sequenced as part of other public health activities and 1970 from the U.S. and elsewhere around the world between 1999 and 2019. The additional Alberta Health isolates were sequenced by the National Microbiology Laboratory (NML)-Public Health Agency of Canada (Winnipeg, Manitoba, Canada) as part of PulseNet Canada activities. Isolates sequenced by the NML for 2018 and 2019 constituted the majority of reported *E. coli* O157:H7 cases for those years (217 of 247; 87.9%). U.S. and global isolates from both cattle and humans were identified from previous literature (n=104) *Franz et al., 2019* and BV-BRC (n=193). As both processed beef and live cattle are frequently imported into Alberta from the U.S., we selected additional *E. coli* O157:H7 sequences available through the U.S. CDC's PulseNet BioProject PRJNA218110. From 2010–2019, 6,791 O157:H7 whole genome sequences were available from the U.S. PulseNet project, 1673 (25%) of which we randomly selected for assembly and clade typing.

This study was approved by the University of Calgary Conjoint Health Research Ethics Board, #REB19-0510. A waiver of consent was granted, and all case data were deidentified.

## Whole genome sequencing, assembly, and initial phylogeny

The 246 isolates for the primary analysis were sequenced using Illumina NovaSeq 6000 and assembled into contigs using the Unicycler v04.9 pipeline, as described previously (BioProject PRJNA870153) (*Bumunang et al., 2022*). Raw read FASTQ files were obtained from Alberta Health for the additional 445 isolates sequenced by the NML and from NCBI for the 152 U.S. and 54 global sequences. We used the SRA Toolkit v3.0.0 to download sequences for U.S. and global isolates using their BioSample (i.e. SAMN) numbers. The corresponding FASTQ files could not be obtained for the six U.S. and seven global isolates we had selected (*Figure 7*).

PopPUNK v2.5.0 was used to cluster Alberta isolates and identify any outside the O157:H7 genomic cluster (*Figure 7—figure supplement 1*; *Lees et al., 2019*). For assembling and quality checking (QC) all sequences, we used the Bactopia v3.0.0 pipeline (*Petit and Read, 2020*). This pipeline performed an initial QC step on the reads using FastQC v0.12.1, which evaluated read count, sequence coverage, and sequence depth, with failed reads excluded from subsequent assembly. None of the isolates were eliminated during this step for low read quality. We used the Shovill v1.1.0 assembler within the Bactopia pipeline to de novo assemble the Unicycler contigs for the primary analysis and raw reads from the supplementary datasets. Trimmomatic was run as part of Shovill to trim adapters and read ends with quality lower than six and discard reads after trimming with overall quality scores lower than 10. Bactopia generated a quality report on the assemblies, which we assessed based on number of contigs (<500), genome size (≥5.1 Mb), N50 (>30,000), and L50 (≤50). Low-quality assemblies were removed. This included one U.S. sequence, for which two FASTQ files had been attached to a single BioSample identifier; the other sequence for the isolate passed all quality checks and remained in the analysis. Additionally, 16 sequences from the primary analysis dataset and four from the extended Alberta data had a total length of <5.1 Mb. These sequences corresponded exactly to those identified by the PopPUNK analysis to be outside the primary *E. coli* O157:H7 genomic cluster (*Figure 7—figure supplement 1*). Finally, although all isolates were believed to be of cattle or clinical origin during the initial selection, a detailed metadata review identified one isolate of environmental origin in the primary analysis dataset and eight that had been isolated from food items in the extended Alberta data. These were excluded. We used STECFinder v1.1.0 (*Zhang et al., 2021*) to determine the Shiga toxin gene (*stx*) profile and confirm the *E. coli* O157:H7 serotype using the *wzy* or *wzx* O157 O-antigen genes and detection of the H7 H-antigen.

Bactopia's Snippy workflow, which incorporates Snippy v4.6.0, Gubbins v3.3.0, and IQTree v2.2.2.7, followed by SNP-Sites v2.5.1, was used to generate a core genome SNP alignment with recombinant blocks removed. The maximum likelihood phylogeny of the core genome SNP alignment generated by IQTree was visualized in Microreact v251. The number of core SNPs between isolates was calculated using PairSNP v0.3.1. Clade was determined based on the presence of at least one defining SNP for the clade as published previously (*Strachan et al., 2015*) Isolates were identified to the clade level, except for clade G where we separated out subclade G(vi).

After processing, we had 229 isolates (121 human, 108 cattle) in our primary sample and 430 additional Alberta Health isolates (*Figure 7*). We had 178 U.S. or global isolates from previous literature (n=88; U.S. n=41, global n=47) and BV-BRC (n=90; U.S. n=75, global n=15). Of the 1673 isolates randomly sampled from the U.S. PulseNet project, 1560 were successfully assembled and passed

**Table 3.** Analyses conducted and model priors.

| Analysis | Isolates included | Isolates remaining after down-sampling | Tree model | Substitution model | Clock model |
|---|---|---|---|---|---|
| Primary analysis* | Alberta 2007–2015 (n=229) | 115 human 84 cattle | Structured coalescent with two demes (1 per species); $N_e$ initial value 10 and Weibull distribution (shape = 1, scale = 100); migration initial value 1.0 and Exponential distribution (mean = 1) | HKY with empirical frequencies and discrete Gamma site model with four categories | Relaxed log-normal with initial value $1.5\times10^{-5}$ and Log-Normal distribution (M=$1.5\times10^{-5}$, S=1.5, with mean in real space) |
| Alberta long-term persistence[†] | Alberta 2007–2019 (n=657) | 274 human 84 cattle | Coalescent constant population with $N_e$ initial value 10 and Weibull distribution (shape = 1, scale = 100) | HKY with empirical frequencies and discrete Gamma site model with four categories | Relaxed log-normal with initial value $1.5\times10^{-4}$ and Log-Normal distribution (M=$1.5\times10^{-4}$, S=1.5, with mean in real space) |
| Global circulation (unstructured)[†] | Alberta 2007–2019 (n=657) The U.S. 1999–2019 (n=492) Global 2007–2016 (n=61) | Alberta: 274 humans, 84 cattle U.S.: 312 humans, 38 cattle Global: 39 humans, 22 cattle | Coalescent constant population with $N_e$ initial value 10 and Weibull distribution (shape = 1, scale = 100) | HKY with empirical frequencies and discrete Gamma site model with four categories | Relaxed log-normal with initial value $1.5\times10^{-4}$ and Log-Normal distribution (M=$1.5\times10^{-4}$, S=1.5, with mean in real space) |
| Global circulation (structured)[†] | Alberta 2007–2019 (n=657) The U.S. 1999–2019 (n=492) Global 2007–2016 (n=61) | Alberta: 274 humans, 84 cattle U.S.: 312 humans, 38 cattle Global: 39 humans, 22 cattle | Structured coalescent with three demes (1 per geography); $N_e$ initial value 10 and Weibull distribution (shape = 1, scale = 100); migration initial value 1.0 and Exponential distribution (mean = 1) | HKY with empirical frequencies and discrete Gamma site model with four categories | Relaxed log-normal with initial value $1.5\times10^{-4}$ and Log-Normal distribution (M=$1.5\times10^{-4}$, S=1.5, with mean in real space) |

*A primary analysis was run using four different random seeds to confirm that all converged to the same solution. The four runs were combined to produce the final maximum clade credibility tree and state transition estimates. Model priors from this analysis were also used in an analysis in which draws were taken from the prior distribution, as opposed to the posterior distribution, to confirm that the final results were not overly influenced by the choice of priors.

[†]Clade A isolates were excluded from these analyses given the very small number available from any locale and clade A's divergence from the rest of the clades.

QC. These included 309 clade G isolates, all of which we included in the analysis; we also randomly sampled and included 69 non-clade G isolates from this sample.

## Phylodynamic and statistical analyses

For our primary analysis, we created a timed phylogeny, a phylogenetic tree on the scale of time, in BEAST2 v2.6.7 using the structured coalescent model in the Mascot v3.0.0 package with demes for cattle and humans (*Table 3*). Sequences were down-sampled prior to analysis if within 0–2 SNPs and <3 m from another sequence from the same host type, leaving 115 human and 84 cattle isolates in the primary analysis (*Table 3*). The analysis was run using four different seeds to confirm that all converged to the same solution, and tree files were combined before generating a maximum clade credibility (MCC) tree. State transitions between cattle and human isolates over the entirety of the tree, with their 95% highest posterior density (HPD) intervals, were also calculated from the combined tree files. We determined the influence of the prior assumptions on the analysis (*Table 3*) with a run that sampled from the prior distribution (Appendix 1). We conducted a sensitivity analysis in which we randomly subsampled 84 of the human isolates so that both species had the same number of isolates in the analysis.

LPLs were identified based on following criteria: (1) a single lineage of the MCC tree with a most recent common ancestor (MRCA) with ≥95% posterior probability; (2) all isolates ≤30 core SNPs from one another; (3) contained at least 1 cattle isolate; (4) contained ≥5 isolates; and (5) the isolates were collected at sampling events (for cattle) or reported (for humans) over a period of at least 1 y. We counted the number of isolates associated with LPLs, including those down-sampled prior to the

phylodynamic analysis. We conducted sensitivity analyses examining different SNP thresholds for the LPL definition.

From non-LPL isolates, we estimated the number of local transient isolates vs. imported isolates. For the 121 human *E. coli* O157:H7 isolates in the primary sample prior to down-sampling, we determined what portion belonged to locally persistent lineages and what portion was likely to be from local transient *E. coli* O157:H7 populations vs. imported. Human isolates within the LPLs were enumerated (n=46). The 75 human isolates outside LPLs included 56 clade G(vi) isolates and 19 non-G(vi) isolates. Based on the MCC tree from the primary analysis, none of the non-G(vi) human isolates were likely to have been closely related to an isolate from the Alberta cattle population, suggesting that all 19 were imported. As a proportion of all non-LPL human isolates, these 19 constituted 25.3%. While it may be possible that all clade G(vi) isolates were part of a local evolving lineage, it is also possible that the exchange of both cattle and food from other locations was causing the regular importation of clade G(vi) strains and infections. Thus, we used the proportion of non-LPL human isolates outside the G(vi) clade to estimate the proportion of non-LPL human isolates within the G(vi) clade that were imported; i.e., $56 \times 25.3\% = 14$. We then conducted a similar exercise for cattle isolates.

To contextualize our results in terms of the ongoing human disease burden, we created a timed phylogeny using a constant, unstructured coalescent model of the 199 Alberta isolates from the primary analysis and the additional Alberta Health isolates (*Figure 7*). The two sets of sequences were combined and down-sampled, leaving 272 human and 84 cattle isolates (*Table 3*). We identified LPLs as above, and leveraged the near-complete sequencing of isolates from 2018 and 2019 to calculate the proportion of reported human cases associated with LPLs.

Finally, we created a timed phylogeny of Alberta, U.S., and global from 1996 to 2019 to test whether the LPLs were linked to ancestors from locations outside Canada (*Table 3*). Due to the size of this tree, we created both unstructured and structured versions. Clade A isolates were excluded due to their small number in Alberta and high level of divergence from other *E. coli* O157:H7 clades. Down-sampling was conducted separately by species and location. The phylogeny included 358 Alberta, 350 U.S., and 61 global isolates after down-sampling.

All BEAST2 analyses were run for 100,000,000 Markov chain Monte Carlo iterations or until all parameters converged with effective sample sizes >200, whichever was longer. Exact binomial 95% confidence intervals (CIs) were computed for proportions.

## Acknowledgements

We would like to acknowledge Dr. Angela Ma, Hannah Tyrrell, and Dr. Surangi Thilakarathna for their work preparing clinical isolates for sequencing, and Dr. Jesse Berman for reviewing an early version of this manuscript. The authors would like to acknowledge and thank the PulseNet participating laboratories, whose data was used for the creation of this publication. Funding for this work was provided by the Beef Cattle Research Council (FOS.01.18). The sponsor had no role in the study design; collection, analysis, or interpretation of data; writing of the report; or the decision to submit the paper for publication.

## Additional information

### Funding

| Funder | Grant reference number | Author |
| --- | --- | --- |
| Beef Cattle Research Council | FOS.01.18 | Gillian AM Tarr<br>Linda Chui<br>Kim Stanford<br>Stephen B Freedman<br>Chad R Laing<br>Tim A McAllister |

The funders had no role in study design, data collection and interpretation, or the decision to submit the work for publication.

## Author contributions
Gillian AM Tarr, Conceptualization, Data curation, Software, Formal analysis, Funding acquisition, Visualization, Methodology, Writing – original draft; Linda Chui, Kim Stanford, Conceptualization, Resources, Supervision, Funding acquisition, Methodology, Writing – review and editing; Emmanuel W Bumunang, Data curation, Investigation, Writing – review and editing; Rahat Zaheer, Investigation, Methodology, Writing – review and editing; Vincent Li, Data curation, Validation, Writing – review and editing; Stephen B Freedman, Conceptualization, Funding acquisition, Project administration, Writing – review and editing; Chad R Laing, Conceptualization, Funding acquisition, Methodology, Writing – review and editing; Tim A McAllister, Conceptualization, Resources, Supervision, Funding acquisition, Methodology, Project administration, Writing – review and editing

## Author ORCIDs
Gillian AM Tarr https://orcid.org/0000-0001-7372-1034

## Ethics
This study was approved by the University of Calgary Conjoint Health Research Ethics Board, #REB19-0510. A waiver of consent was granted, and all case data were deidentified.

Reviewer #1 (Public review): https://doi.org/10.7554/eLife.97643.3.sa1
Author response https://doi.org/10.7554/eLife.97643.3.sa2

## Additional files

### Supplementary files
MDAR checklist

Source data 1. List of accession numbers for sequences used with deidentified metadata.

### Data availability
Sequencing data have been deposited in SRA. Accession numbers and deidentified metadata for all sequences used, including existing sequences and those sequenced for this study, are listed in the Source Data.

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

## Appendix 1

### STEX case definition

Alberta Health defines a confirmed case of STEC, including *E. coli* O157:H7, as STEC isolation or Shiga toxin antigen or nucleic acid detection. Clinical illness, which may include diarrhea, bloody diarrhea, abdominal cramps, hemolytic uremic syndrome, thrombocytopenia purpura, or pulmonary edema, may or may not be present.

### Sampling from the prior distribution

Results in our Bayesian phylodynamic analyses are drawn from posterior distributions, which are influenced by both the data and the prior information we have about the system (*Table 3*). In order to confirm that our primary results were not overly influenced by our prior assumptions, we conducted an analysis in which the sampling draws were made from the prior distribution, as opposed to the posterior distribution. We graphed these results against the sampling draws made from the posterior distributions from the four runs conducted for our primary analysis (each performed with a different random seed). The comparison shows that the draws from the prior distribution differ markedly from the draws from the posterior distributions for the model's key parameters (*Figure 3—figure supplement 1*). From this, we concluded that our prior assumptions were not overly influencing the results of the primary analysis.

