## [Editor Report · eLife Assessment]

This **valuable** study revealed numerous distinct lineages that evolved within a local human population in Alberta, Canada, leading to persistent cases of *E. coli* O157:H7 infections for over a decade and highlighting the ongoing involvement of local cattle in disease transmission, as well as the possibility of intermediate hosts and environmental reservoirs. This study also showed a shift towards more virulent stx2a-only strains becoming predominant in the local lineages. The evidence supporting the role played by cattle in the transmission system of human cases of *E. coli* O157:H7 in Alberta is **solid**.

---

## [Referee Report · Reviewer #1 (Public review)]

Summary:

This is a high-quality, well-thought through analysis of STEC transmission in Alberta, Canada.

Strengths:

* The combined human and animal sampling is a great foundation for this kind of study.

* Phylogenetic analyses seem to have been carried out in a high quality fashion.

Comments on the revised version:

I'd like to thank the authors for the diligence with which they addressed my comments. I agree with their points and am happy for the manuscript to proceed.

---

## [Author Response]

The following is the authors’ response to the original reviews.

**Reviewer #1 (Public Review):**
Summary:A nice study trying to identify the relationship between *E. coli* O157 from cattle and humans in Alberta, Canada.Strengths:(1) The combined human and animal sampling is a great foundation for this kind of study.(2) Phylogenetic analyses seem to have been carried out in a high-quality fashion.Weaknesses:I think there may be a problem with the selection of the isolates for the primary analysis. This is what I'm thinking:(1) Transmission analyses are strongly influenced by the sampling frame.(2) While the authors have randomly selected from their isolate collections, which is fine, the collections themselves are not random.(3) The animal isolates are likely to represent a broad swathe of diversity, because of the structured sampling of animal reservoirs undertaken (as I understand it).(4) The human isolates are all from clinical cases. Clinical cases of the disease are likely to be closely related to other clinical cases, because of outbreaks (either detected, or undetected), and the high ascertainment rate for serious infections.(5) Therefore, taking an equivalent number of animal and clinical isolates, will underestimate the total diversity in the clinical isolates because the sampling of the clinical isolates is less "independent" (in the statistical sense) than sampling from the animal isolates.(6) This could lead to over-estimating of transmission from cattle to humans.

We appreciate the reviewer’s careful thoughts about our sampling strategy. We agree with points (1) and (2), and we have provided additional details on the animal collections as requested (lines 95-101).

We agree with point (3) in theory but not in fact. As shown in Figure 3, the cattle isolates were very closely related, despite the temporal and geographic breadth of sampling within Alberta. The median SNP distance between cattle sequences was 45 (IQR 36-56), compared to 54 (IQR 43-229) SNPs between human sequences from cases in Alberta during the same years. Additionally, as shown in Figure 2, only clade A and B isolates – clades that diverge substantially from the rest of the tree – were dominated by human cases in Alberta. We have better highlight this evidence in the revision (lines 234-236 and 247-249).

We agree with the reviewer in point (4) that outbreaks can be an important confounder of phylogenetic inference. This is why we down-sampled outbreaks (based on genetic relatedness, not external designation) in our extended analyses. We did not do this in the primary analysis, because there were no large clusters of identical isolates. Figure 3b shows a limited number of small clusters; however, clustered cattle isolates outnumbered clustered human isolates, suggesting that any bias would be in the opposite direction the reviewer suggests. In the revision, we down-sampled all analyses and, indeed, the proportion of human lineages descending from cattle lineages increased (lines 259-261). Regarding severe cases being oversampled among the clinical isolates, this is absolutely true and a limitation of all studies utilizing public health reporting data. We made this limitation to generalizability clearer in the discussion. However, as noted above, clinical isolates were more variable than cattle isolates, so it does not appear to have heavily biased the analysis (lines 490-495).

We disagree with the reviewer on point (5). While the bias toward severe cases could make the human isolates less independent, the relative sampling proportions are likely to induce greater distance between clinical isolates than cattle isolates, which is exactly what we observe (see response to point (3) above). Cattle are *E. coli* O157:H7’s primary reservoir, and humans are incidental hosts not able to sustain infection chains long-term. Not only is the bacteria prevalent among cattle, cattle are also highly prevalent in Alberta. Thus, even with 89 sampling points, we are still capturing a small proportion of the *E. coli* O157:H7 in the province. Being able to sample only a small proportion of cattle’s *E. coli* O157:H7 increases the likelihood of only sampling from the center of the distribution, making extreme cases such as that shown at the very bottom of the tree in Figure 4, rare and important. In comparison, sampling from human cases constitutes a higher proportion of human infections relative to cattle, and is therefore more representative of the underlying distribution, including extremes. We added this point to the limitations (lines 495-504). As with the clustering above, if anything, this outcome would have biased the study away from identifying cattle as the primary reservoir. Additionally, the relatively small proportion of cattle sampled makes our finding that 15.7% of clinical isolates were within 5 SNPs of a cattle isolate, the distance most commonly used to indicate transmission for *E. coli* O157:H7, all the more remarkable.

Because of the aforementioned points, we disagree with the reviewer’s conclusion in point (6). If a bias exists, we believe transmission from cattle-to-humans is likely underestimated for the reasons given above. Not only do all prior studies indicate ruminants as the primary reservoirs of *E. coli* O157:H7, and humans as only incidental hosts, our specific data do not support the reviewer’s individual contentions. The results of the sensitivity analysis the reviewer recommended is consistent with the points we outlined above, estimating that 94.3% of human lineages arose from cattle lineages (vs. 88.5% in the primary analysis). We have opted to retain the more conservative estimate of the primary analysis, which includes a more representative number of clinical cases.

(7) We hypothesize that the large proportion of disease associated with local transmission systems is a principal cause of Alberta's high *E. coli* O157:H7 incidence" - this seems a bit tautological. There is a lot of O157 because there's a lot of transmission. What part of the fact it is local means that it is a principal cause of high incidence? It seems that they've observed a high rate of local transmission, but the reasons for this are not apparent, and hence the cause of Alberta's incidence is not apparent. Would a better conclusion not be that "X% of STEC in Alberta is the result of transmission of local variants"? And then, this poses a question for future epi studies of what the transmission pathway is.

The reviewer is correct, and the suggestion for the direction of future studies was our intent with this statement. We have removed this sentence.

**Reviewer #1 (Recommendations For The Authors):**
(1) To address my concerns about the different sampling frames in humans and animals, I would suggest a sensitivity analysis, using something like the following strategy. Make a phylogeny of all the available genome sequences from humans and cattle from Alberta. Phylogenetically sub-sample the tree, using something like Treemer (https://github.com/fmenardo/Treemmer), to remove phylogenetically redundant isolates from the same host type. Randomly select 100 human and 100 animal isolates from this non-redundant tree, and re-do your analysis.

Although we originally down-sampled outbreaks for our analysis of the extended Alberta tree (2007-2019), we had not done this systematically for all analyses. We were not able to use the recommended Treemer tool, because we did not see a way to incorporate the timing of sequences. Because the objective of our study was to evaluate persistence, we did not want to exclude identical sequences that were separated in time and thus could be indicating persistence. To accomplish this, we developed a utility that allowed us to incorporate the temporality of sequences. Using this utility, we systematically down-sampled all sequences that met the following conditions: (1) within 0-2 SNPs of another sequence and (2) no gaps in sequence set >2 months. The second condition means that for any set of sequences within 0-2 SNPs of one another, there can be no more than 2 months without a sequence from the set. Similar sequences that occur beyond this 2-month-cutoff would be considered a separate set for down-sampling. This cutoff was chosen based on the epidemiology of *E. coli* O157 outbreaks, which are generally either point-source or continuous-source outbreaks. Intermittent outbreaks of a single strain are believed to arise from distinct contamination events and are exactly the type of phenomena we are seeking to identify. We have added details on down-sampling to the Methods (lines 178-180).

After down-sampling, our primary analysis included 115 human and 84 cattle isolates. T conduct the recommended sensitivity analysis, we further randomly subsampled the human isolates, selecting 84 to match the number of cattle isolates. As we suggested in our initial response, and contrary to the reviewer’s concern, subsampling in this way accentuated the results, with 94.3% of human lineages inferred as arising from cattle lineages, compared to 88.5% in the primary analysis. This sensitivity analysis also identified 10 of the 11 LPLs identified in the primary analysis. The LPL not identified had 5 isolates in the primary analysis, the minimum for definition as an LPL, and was reduced to 4 isolates through subsampling. This sensitivity analysis is shown in Suppl. Figure S3.

(2) This is the first time I've seen target diagrams used for SNP distances, I'm not sure of their value compared with histograms. They seem to emphasise the maximum distance, rather than the largest number of isolates. I.e. most isolates are closely related, but the diagram emphasises the small number of divergent ones.

In using the target diagrams, we sought to emphasize the bimodal distribution of human-to-closest-cattle SNP differences. However, this is still mostly visible in a histogram, so we have replaced the target diagrams with a histogram as suggested (Figure 3).

(3) L130 - fastqc doesn't trim adapters and read ends, there will be something else like trimmomatic which does.

The reviewer is correct, and we appreciate them catching this error. Trimmomatic is incorporated into the Shovill pipeline, which was the assembler we used through the Bactopia pipeline. We have updated the Methods to indicate this (lines 142-144).

(4) I find the flow of the article a bit confusing. You have your primary analysis, but Figure 2, which is a secondary analysis, comes before Figure 3. Which is the primary analysis? For me, primary analysis results should come first, or at least signpost a bit better.

Figure 2 is not a secondary analysis. It is intended to provide an overview of the isolates used from the phylogenetic perspective, just as the diagram in Figure 1 provides an overview of the isolates by analysis. The secondary analyses are shown in Figures 5-7. We have added a sub-header, “Description of Isolates”, to the section referring to Figure 2, to clarify (line 232).

(5) Locally persistent lineage definition. What is the rationale for the different criteria signifying locally persistent lineages? There is nothing in some of your criteria e.g. all isolates <30 SNPs from each other, which indicates that it is locally persistent - could have been transmitted to Japan (just to pick a place at random), causing a bunch of cases there, and then come back for all we know. Would that be a locally persistent lineage? Did you use the MCC tree here? That is a sub-sample of your full dataset, I am not sure what exactly you're trying to say with the LPLs, but maybe using a larger dataset would be better? Also, there are lots of STEC genomes available from e.g. UK and USA, by only including a fraction of these, you limit the strength of the inferences you can make about locally persistent lineages unless you know that they don't see the G sub-lineage that you observe.

The reviewer raises multiple points here. First, regarding our definition of LPLs, it is intended to identify those lineages that pose a threat to populations in the specific geographic area (“local”) for at least 1 year (“persistent”) that are likely to be harbored in local reservoirs. Each of the criteria contributes to this definition.

(1) A single lineage of the MCC tree with a most recent common ancestor (MRCA) with ≥95% posterior probability: This criterion provides confidence in the given isolates being part of a single, defined lineage. The posterior probability gives the probability that the topology of the tree is accurate, based on the data provided and the chosen model of evolution. In other words, we required at least 95% probability that the lineage was correct, and in practice the posterior probability of the lineages we defined as LPLs was 99.7-100% (we have added this detail to the text, lines 269-270). We also added a sensitivity analysis, shown in Suppl. Figure S4, which shows all sampled trees. We find that the essential structure of the tree around the LPLs we defined is well-supported.

(2) All isolates ≤30 core SNPs from one another: This criterion limited LPLs to those lineages where the isolates were closely related. We did not want to limit LPLs to those that might define an outbreak, for example using a 5-10 SNP threshold, because the point of the study is to identify lineages that persistently cause disease over longer periods than a normal outbreak. Pathogens evolve over time in their reservoirs, leading to greater SNP distances, and we wanted to allow for this. The U.S. CDC has acknowledged a similar concern for such persistent lineages in its definition of REP strains, which it has defined based on ranges of 13-104 allele differences by cgMLST. Thus, our choice of 30 core SNPs as the threshold is in line with current practice in the emerging science on persistence of enteric pathogens. We have also added a sensitivity analysis examining alternate SNP thresholds, shown in Suppl. Figure S5, which results in clusters of LPLs identified in the primary analysis being grouped into larger lineages. Additionally, in the tree showing our primary analysis (Figure 4), we now note the minimum number of SNPs all isolates within the lineage differ by.

(3) *Contained at least 1 cattle isolate:* This criterion increases confidence that the lineage is indeed “local”. Unlike humans, cattle are not known to be routinely infected by imported food products, and they do not make roundtrip journeys to other locations, as humans infected during travel do. Cattle themselves may be imported into Alberta while infected, and cattle in Alberta can be infected by other imported animals. In these cases, if the STEC strains the cattle harbor persist for ≥1 year, they become the type of lineages we are interested in as LPLs, regardless where they previously came from, because they are now potential persistent sources of infection in Alberta. By including at least one cattle isolate in each LPL, the only way an identified LPL is not actually local is if cattle are imported from the lineage’s reservoir community elsewhere (e.g., in Japan, as the reviewer suggested), the lineage is persisting in that non-Alberta reservoir, and newly infected cattle are imported repeatedly over 1 or more years. This could feasibly explain G(vi)-AB LPL 5 (Figure 4), which is entirely composed of cattle. Indeed, such an explanation would be consistent with the lack of new cases from this LPL after 2015 in the extended analysis (Figure 5). However, for all other LPLs, which contain both cattle and human isolates, for the LPL to not be local, both cattle and human cases would have to be imported from the same non-Alberta reservoir. While this is possible, the probability of such a scenario is low, and it decreases the more isolates are in an LPL. For the average LPL, this means 4 human and 6 cattle cases would need to be imported from a non-Alberta reservoir over several years. Given that our study is only a random sample of the total STEC cases and cattle in Alberta from 2007-2015, these numbers are underestimates of the true absolute number of cases and cattle associated with LPLs that would have to be explained by importation if the LPL were not local. We have added some explanation of the possibility of importation in the Discussion where we discuss the LPL criteria (lines 376-380).

(4) Contained ≥5 isolates: In concert with criterion 3, this criterion guards against anomalies being counted as LPLs. By requiring at least 5 isolates in an LPL after down-sampling, at least 5 infection events must have occurred from the LPL, reducing the likelihood of importation explaining the LPL and emphasizing more significant LPLs.

(5) The isolates were collected at sampling events (for cattle) or reported (for humans) over a period of at least 1 year: This criterion defines the persistence aspect of the LPL. In the primary analysis, the LPLs we identified persisted for an average of 8 years, with the shortest persisting for 5 years (these details have been added to the text, lines 268-269). Incorporating the extended analysis, several LPLs persisted for the full 13 years of the study.

Regarding using additional non-Alberta isolates to help rule out importation, we have expanded the number of U.S. and global isolates included in the importation analysis, over-sampling clade G isolates from the U.S. (Figure 7). As cattle trade is substantially more common with the U.S. than other countries, we felt it most important to focus on the U.S. as a potential source of both imported cattle and human cases. Our results from this analysis show that only 9 of 494 (1.8%) U.S. isolates occurred in the LPLs we defined in the primary analysis, and all occurred after Alberta isolates (lines 313-317). Although we also added more global isolates, we still found that none were associated with the Alberta LPLs.

(6) Given the importance of sampling for a study like this, some more information on animal sampling studies should be included here.

We have added details on the cattle sampling to the Methods (lines 95-101).

(7) L172 - do you mean an MRCA with >- 95% probability of location in Alberta?

Location in Alberta was not determined from the primary analysis, which defined the LPLs, as only Alberta isolates were included in that analysis. As described above, this criterion meant that we required at least 95% probability that the tree topology at the lineage’s MRCA was correct, and in practice the posterior probability of the lineages we defined as LPLs was 99.7-100%.

(8) Need a supplementary figure of just clade G from Figure 2.

We have added a sub-tree diagram of clade G(vi) as Figure 2b.

**Reviewer #2 (Public Review):**
This study identified multiple locally evolving lineages transmitted between cattle and humans persistently associated with *E. coli* O157:H7 illnesses for up to 13 years. Furthermore, this study mentions a dramatic shift in the local persistent lineages toward strains with the more virulent stx2a-only profile. The authors hypothesized that this phenomenon is the large proportion of disease associated with local transmission systems is a principal cause of Alberta's high *E. coli* O157:H7 incidence. These opinions more effectively explain the role of the cattle reservoir in the dynamics of *E. coli* O157:H7 human infections.(1) The authors acknowledge the possibility of intermediate hosts or environmental reservoirs playing a role in transmission. Further discussion on the potential roles of other animal species commonly found in Alberta (e.g., sheep, goats, swine) could enhance the understanding of the transmission dynamics. Were isolates from these species available for analysis? If not, the authors should clearly state this limitation.”

We have expanded the discussion of other species in Alberta, as suggested, including other livestock, wildlife, and the potential role of birds and flies (lines 353-360). Unfortunately, we did not have sequences available from other species, which we have added to the limitations (lines 487-490).

(2) The focus on *E. coli* O157:H7 is understandable given its prominence in Alberta and the availability of historical data. However, a brief discussion on the potential applicability of the findings to non-O157 STEC serogroups, and the limitations therein, would be beneficial. Are there reasons to believe the transmission dynamics would be similar or different for other serogroups?

We appreciate this comment and have expanded our discussion of relevance to non-O157 STEC (lines 452-460). Other authors have proposed that transmission dynamics differ, and studies of STEC risk factors, including our own, support this. However, there has been very little direct study of non-O157 transmission dynamics and there is even less cross-species genomic and metadata available for non-O157 isolates of concern.

(3) The authors briefly mention the need for elucidating local transmission systems to inform management strategies. A more detailed discussion on specific public health interventions that could be targeted at the identified LPLs and their potential reservoirs would strengthen the paper's impact.

We agree with the reviewer that this would be a good addition to the manuscript. The public health implications for control are several and extend to non-STEC reportable zoonotic enteric infections, such as Campylobacter and *Salmonella*. We have added a discussion of these (lines 460-465, 467-485).

(4) Understanding the relationship between specific risk factors and *E. coli* O157:H7 infections is essential for developing effective prevention strategies. Have case-control or cohort studies been conducted to assess the correlation between identified risk factors and the incidence of *E*. *coli* O157:H7 infections? What methodologies were employed to control for potential confounders in these studies?

Yes, there have been several case-control studies of reported cases. Many of these are referenced in the discussion in terms of the contribution of different sources to infection. As risk factors were not the focus of the current study, we believe a thorough discussion of the literature on the aspects of these various studies is beyond our scope. However, we have added some details on the risk factors themselves (lines 72-79).

(5) The study's findings are noteworthy, particularly in the context of *E. coli* O157:H7 epidemiology. However, the extent to which these results can be replicated across different temporal and geographical settings remains an open question. It would be constructive for the authors to provide additional data that demonstrate the replication of their sampling and sequencing experiments under varied conditions. This would address concerns regarding the specificity of the observed patterns to the initial study's parameters.

We appreciate the reviewer’s comment, as we are currently building on this analysis with an American dataset with different types of data available than were used in this study. Aligned with this work, we have added a comment on the adaptation of our method to other settings with different types of data (lines 448-450). We also added a sensitivity analysis to the manuscript simulating a different sampling approach (Suppl. Fig. S3), which should be informative to this question.

**Reviewer #2 (Recommendations For The Authors):**
Minor comments.(1) Figure 1: The figure is a critical visual representation of the study's findings and should be given prominent emphasis. It is essential that the key discoveries of the research are clearly depicted and explained in this visual format. The authors should ensure that Figure 1 is detailed and informative enough to stand out as a central piece of the study.

Figure 1 is the diagram of sample numbers, locations, and corresponding analyses. We assume that the reviewer means to refer to Figure 2. Although the inclusion of >1,200 isolates makes the tree difficult to see in detail, we have made some modifications to make the findings clearer. First, we changed the clade coloration such that the only subclade differentiated is G(vi). We have removed the stx metadata ring to focus attention on the location and species of the isolates, as stx data are described in Table 1. Finally, we have added a sub-tree diagram of clade G(vi), colored by location. This makes clear the large sections of the subclade dominated by isolates from one location or another, and the limited areas where they overlap.

(2) Figures 2 and 4: While these figures contribute to the presentation of the data, they appear to be somewhat rudimentary in their current form. The lack of detailed annotations regarding the clustering of different strains is a notable omission. I recommend that the authors refine these figures to include comprehensive labeling that clearly delineates the various bacterial clusters. Enhanced graphical representation with clear annotations will aid readers in better understanding the study's findings.

We appreciate this suggestion. We have remade all trees generated by the BEAST 2 analyses in R, rather than FigTree. This has allowed us to annotate the trees with additional information on the LPLs and we believe provides a clearer picture of each LPL.

(3) Supplemental Table S1: The supplemental tables are an excellent opportunity to showcase additional data and findings that support the study's conclusions. For Supplemental Table S1, it is recommended that the authors highlight the innovative aspects or novel discoveries presented in this table.

Suppl. Table S1 shows the modeling specifications and priors used in the analyses. These decisions were not in and of themselves novel. The innovation in our methods is due to the development of the LPLs based on the trees resulting from the analyses detailed in Suppl. Table S1, as well as from the application of these models to *E. coli* O157:H7 for the first time. However, we understand the reviewers point and have emphasized the importance of the results shown in Suppl. Table S2 (lines 391-395).

(4) Line 35: "We assessed the role of persistent cross-species transmission systems in Alberta's E. coli O157:H7 epidemiology." change to "We assessed the impact of persistent cross-species transmission systems on the epidemiology of *E. coli* O157:H7 in Alberta."

We have made this change.

(5) To facilitate a deeper understanding of the core findings of the manuscript and to enable the development of effective response strategies, I suggest that the authors provide more information regarding the sequencing data used in the study. This information should at least include aspects such as data accessibility and quality control measures.

We have included a Supplemental Data File that lists all isolates used in the analysis, and the QC measures are detailed in the Methods.